# Three-Dimensional Customized Imaging Reconstruction for Urological Surgery: Diffusion and Role in Real-Life Practice from an International Survey

**DOI:** 10.3390/jpm13101435

**Published:** 2023-09-26

**Authors:** Maria Chiara Sighinolfi, Aurus Dourado Menezes, Vipul Patel, Marcio Moschovas, Simone Assumma, Tommaso Calcagnile, Enrico Panio, Mattia Sangalli, Filippo Turri, Luca Sarchi, Salvatore Micali, Virginia Varca, Filippo Annino, Costantino Leonardo, Giorgio Bozzini, Giovanni Cacciamani, Andrea Gregori, Elena Morini, Stefano Terzoni, Ahmed Eissa, Bernardo Rocco

**Affiliations:** 1Urologic Unit, ASST Santi Paolo e Carlo, La Statale University, 20122 Milan, Italy; 2Clinica da Urologia e Cirurgia Urologica, Teresina 64000-000, Brazil; 3Global Robotic Institute, AdventHealth, Orlando, FL 34747, USA; 4Urologic Unit, Azienda Ospedaliera Universitaria di Modena, 41124 Modena, Italy; 5Urologic Unit, ASST Rhodense—Presidio di Garbagnate Milanese, 20024 Milan, Italy; 6Urologic Unit, San Donato Hospital, 52100 Arezzo, Italy; 7Urologic Unit, IFO Istituto Tumori Regina Elena, 00144 Roma, Italy; 8Urologic Unit, ASST Lariana, 22100 Como, Italy; 9USC Institute of Urology, University of Southern California, Los Angeles, CA 90089, USA; 10Urologic Unit, Fatebenefratelli-Sacco Hospital, 20157 Milan, Italy; 11Urologic Department, Faculty of Medicine, Tanta University, Tanta 31527, Egypt

**Keywords:** three-dimensional (3D) imaging reconstruction, partial nephrectomy, radical prostatectomy, surgical planning, urological practice

## Abstract

Despite the arising interest in three-dimensional (3D) reconstruction models from 2D imaging, their diffusion and perception among urologists have been scarcely explored. The aim of the study is to report the results of an international survey investigating the use of such tools among urologists of different backgrounds and origins. Beyond demographics, the survey explored the degree to which 3D models are perceived to improve surgical outcomes, the procedures mostly making use of them, the settings in which those tools are mostly applied, the surgical steps benefiting from 3D reconstructions and future perspectives of improvement. One hundred responders fully completed the survey. All levels of expertise were allowed; more than half (53%) were first surgeons, and 59% had already completed their training. Their main application was partial nephrectomy (85%), followed by radical nephrectomy and radical prostatectomy. Three-dimensional models are mostly used for preoperative planning (75%), intraoperative consultation and tailoring. More than half recognized that 3D models may highly improve surgical outcomes. Despite their recognized usefulness, 77% of responders use 3D models in less than 25% of their major operations due to costs or the extra time taken to perform the reconstruction. Technical improvements and a higher availability of the 3D models will further increase their role in surgical and clinical daily practice.

## 1. Introduction

The concept of precision surgery has widely entered the urologic field. Urology encompasses interventions whose technique may differ according to individual features. These include not only the patient’s characteristics, which may vary and prompt different approaches, but also disease features (stage and invasion of closer structures) that should be preoperatively known to plan the most accurate and tailored strategy.

In urology, prostate and kidney cancer surgery require a customized approach.

Radical prostatectomy (RP) represents a recommended treatment for intermediate-risk organ-confined prostate cancer; nowadays, indications of RP are also extending to high-risk and locally advanced diseases inside a multimodal therapeutic pathway [1]. One of the biggest challenges of prostate surgery is the tradeoff between the complete removal of the tumor and the preservation of nerve structures responsible for potency, which run close to the gland. The preservation of neurovascular bundles (NVB) is crucial to improve erection recovery, but may result in an incomplete resection of the adjacent tumor and thus lead to a positive surgical margin. This is likely to occur especially in cases of extracapsular extension of cancer (ECE) [2]. The preoperative knowledge of tumor location, burden and capsular distance is of paramount importance to plan the surgical strategy and to optimize both the oncological and functional outcomes of RP. Beyond the prediction of ECE risk, the visual localization of the cancerous lesion inside the prostate could be useful for the surgeon to guide the dissection. This is critical for robotic surgery; the approach mostly takes advantage of preoperative knowledge of local staging given its intrinsic precision derived from magnification and 3D visualization.

The other urologic surgery relying on a tailored strategy is the management of kidney cancer. During the last decades, the use of kidney sparing techniques has gained approval; partial nephrectomy (PN) is currently the recommended approach for T1a renal cancer and should also be pursued for T1b stages whenever technically feasible. The advantages of the approach are connected to the preservation of the kidney unit, with sustained renal function and benefits in terms of cardiovascular outcomes too [3,4]. As for prostate surgery, the introduction of robotics enhanced the diffusion of PN, given its precision in dissecting and suturing, crucial tasks during such a time-dependent surgery as PN [5].

The surgical complexity of PN is strictly connected to the complexity of the renal mass. As more endophytic the lesion is, closer to the hilum or to the collecting system, the higher the difficulty of PN could be. The cancer staging system does not account for all possible variables and is not able to depict an individual’s anatomy [6]. To overcome this issue, nephrometric scores have been developed. Such models merge several characteristics of the mass (i.e., size, exophytic/endophitic ratio, location, rim, and proximity to the hilum) and predict the complication rate, risk of conversion to nephrectomy, and need for—or duration of—ischemia [7,8].

Despite the importance of those scores to forecast surgical complexity, one of the most important surgeon’s needs remains the cognitive “visualization” of the mass inside the kidney, its relationship with relevant structures and possible risks. Tumor diagnosis conventionally occurs with contrast-enhanced computed tomography (CT), which has proven adequate discrimination to characterize renal lesions; magnetic resonance imaging (MRI) represents an alternative to CT that displays a sensitivity and specificity of 86% and 78%, respectively, in distinguishing between malignant and non-malignant features [9,10,11]. Despite the accuracy of such diagnostic tools, the cognitive load demanded of surgeons to reconstruct kidney anatomy could be bothersome, and, thus, conventional imaging modalities are considered suboptimal to guide kidney intervention in surgical practice [12].

During the last decade, three-dimensional (3D) reconstruction from 2D cross-sectional imaging has been given widespread attention and gained popularity among the urological scientific community [13,14,15,16]. Three-dimensional models embody the concept of personalized precision surgery [11], since they are derived from individual features and developed to tailor the intervention to the singular patient. The 3D virtual models provide the surgeon with a better understanding of the surgical anatomy of each case and also an opportunity to highlight anatomical details of interest. Conventional CT and MRI scans depict every element of the patient’s anatomy captured during the imaging process, including those irrelevant to surgery. Three-dimensional models allow the surgeon to focus on details of interest, especially those important to guide a tailored strategy [12].

The production of those models usually involves a team of radiologists, urologists, and bioengineers; once a high-quality 3D virtual model from 2D imaging is created, it can be applied in three different settings, namely the cognitive navigation of the virtual model, its printed version, or in augmented reality (AR) procedures [13,14,15,16,17].

The cognitive 3D reconstruction of imaging has gained a certain popularity, especially for kidney and prostate cancer surgery. Reconstructed models can be displayed and navigated on the smartphone, on the PC desktop or also visualized inside the Da Vinci Tile Pro. Several systems are currently available on the market, with worldwide diffusion, having different costs and different performances.

Despite the rising interest in those models and the number of publications on the topic, their real application in urological practice has been scarcely explored. The aim of this study is to evaluate the perceived role of 3D imaging reconstruction among urologists of different backgrounds and origins through an international survey. The dissemination of 3D models into urological practice is addressed, and possible concerns are unveiled.

## 2. Materials and Methods

This was a cross-sectional study evaluating the role of 3D imaging reconstruction in real-life urological practice through a survey. The survey was developed by two institutions (ASST Santi Paolo e Carlo, University of Milan and Clinica da Urologia e Cirurgia Urologica, Teresina, Brazil). Participants were recruited among Italian residency programs, internationally recognized groups or societies endorsing minimally invasive surgery (Global Robotic Institute, Celebration, FL 34747, USA; Society of Robotic Surgery, East Dundee, IL 60118, USA; AGILE group—Italian Group for advanced laparo-endoscopic and robotic urologic surgery). An inclusion criterion was the previous or current exposure to 3D models for kidney and/or prostate imaging reconstruction. Users were included to provide opinions about the technology. Participants were recruited on a voluntary basis. Responders were given the opportunity to be openly acknowledged in the event of the publication of the outcomes. No fee was offered for participating.

### 2.1. Survey

After a section collecting demographic data, the survey consisted of general questions addressing the following items: (1) to quantify the degree to which 3D models are perceived to improve surgical outcomes; (2) to evaluate the procedures mostly taking advantage of 3D imaging; (3) to identify the settings in which 3D models are mostly applied; (4) to evaluate surgical steps benefiting from 3D models; (5) to address how 3D models could be improved. The DocDo platform (https://www.docdo.com.br) was mentioned as a reference in the text; however, surveys from participants using other systems for 3D reconstruction were included as well.

### 2.2. Full Text of the Survey

Surname, name and affiliation (in case of public acknowledgement)Are you working in an Academic or in a Non-Academic institution?How old are you?How long have you been working as a Urologist?Do you mainly practice?Are you a trainee? How many major laparoscopic/robotic/open surgeries are performedDo you mainly perform major laparoscopic/robotic surgery as?How often do you use 3D imaging reconstruction in your surgicalFor what sort of laparoscopic/robotic/open surgery do you usually use 3D imaging reconstruction? SpecifyDid you experience 3D imaging systems other than DocDo?In which setting do you think that 3D imaging reconstruction is more useful?How many times do you show your patients 3D images to explain the intervention?Do you feel 3D findings may change surgical pre-planning?Kidney surgery: do you feel 3D imaging is more useful to assess: nephrometric score based on 3D; pedicle and vascular anatomy; The volume of the remaining kidney; other comment)Prostate surgery: Do you feel that 3D imaging carries advantages over the 2D for the assessment of: Localization of the main tumor; lesion proximity to the capsule; otherHow much do you believe 3D imaging use is able to improve your surgical outcomes? (5-Likert scale)How can the 3D imaging reconstruction software be improved?

### 2.3. Timeframe

On 21 August 2021, the survey was sent to 600 urologists or urology residents with current or previous known availability of 3D cognitive models. The survey was sent in 3 rounds to reach a preplanned number of 100 responders, and recruitment ended on 1 January 2022. All backgrounds (academic/not academic), settings (robot/lap/open) and levels of expertise were allowed. The survey was built using Google Forms. All results were collected in Google Sheets and exported into Excel. The survey was developed according to the Checklist for Reporting Results of Internet E-surveys (CHERRIES), which is part of the EQUATOR network (Enhancing the QUAlity and Transparency Of health Research) [18].

### 2.4. Participants Selection

Participants were recruited on a voluntary basis without any incentives; they were told upfront about the length of the time of the survey, and the principal investigators were clearly named (MCS, AD, SM and BR) together with their contacts. No personal information nor patient-related data were collected; thus, the survey was not password-protected and was shared only between investigators. Responders were able to review and change their answers (through an “Erase” button). The “view rate” was not available. No statistical correction was applied, given the lack of need to adjust for nonrepresentative samples. Several questions were answered on a 5-point Likert scale, in which 5 was “very highly impacting or useful”, 4 was “highly impacting or useful”, 3 was “Impacting or useful”, 2 was “low impacting or useful” and 1 was “not impacting at all/useless” for specific purposes.

### 2.5. Statistics

Statistical analysis was performed with SAS^®^9 (SAS Inc., Cary, NC 27513, USA). Categorical variables were analyzed as frequencies. The internal consistency of the questionnaire was assessed by the calculation of Cronbach’s alpha coefficient. The significance threshold for all calculations was 5%. Categorical variables were analyzed as frequencies and compared with the chi-square test (Fisher’s exact test if expected frequencies in the contingency tables were <5).

## 3. Results

Out of the 600 invited, 100 completed the survey (16.6%). Each questionnaire was answered once by each single user.

Cronbach’s alpha was satisfactory (0.839) even after the removal of every single item [range 0.804–0.854], thus suggesting good reliability of the questionnaire.

### 3.1. Demographics

Responders’ countries of origin were Italy, Brazil, the United States, Egypt, and Portugal. Overall, 55 robotic, 25 laparoscopic and 9 open surgeons were included; the remaining (11) did not define themselves as belonging to a certain category. Forty-six (46%) have been working as urologists for less than 10 years, 35% from 10 to 20 years, and 19% had more than 20 years of urological expertise. More than half (53%) mostly worked as first surgeons, and 59% had already completed their training. Seventy (70%) worked inside medium- to high-volume institutions (>100 major procedures/year). Forty-six participants (46%) had used more than two platforms for 3D imaging reconstruction; 14 also had experience using 3D-printed models. If C-alpha is calculated as stated in the methods, the data are not reported here.

### 3.2. Outcomes

The main application of 3D cognitive models is partial nephrectomy (85%), followed by radical nephrectomy and radical prostatectomy; some responders use 3D models for radical cystectomy, female pelvic surgery, complex stone and UPJ surgery as well. By using a 5-point Likert scale, the settings in which 3D cognitive models are mostly used are preoperative planning (75% responders found models very/highly useful) (Figure 1) and intraoperative consultation and tailoring (76% responders found models very/highly useful) (Figure 2).

Forty-seven (47%) defined models as very/highly useful for educational purposes, whereas only 33% found them very/highly useful for patients’ counseling. In kidney surgery, the main application was the evaluation of pedicle and vascular anatomy rather than tumor localization; in prostate surgery, 70% found 3D models useful to assess lesion proximity to the capsule. More than half recognized that 3D models may improve surgical outcomes to a very high/high degree (Figure 3).

The majority (72%) only marginally uses 3D models to improve patient–doctor communication (images shown to the patient in less than 25% of the cases). This is consistent with the fact that only 33% find models highly/very highly useful for patients’ counseling.

The offline availability of 3D models, the incorporation of nephrometric scores and the presence of a ruler are suggestions to improve 3D imaging, as reported by 41%, 28% and 26% of responders, respectively.

Despite their recognized usefulness, the majority of responders (77%) use 3D models in less than 25% of their major robotic, laparoscopic or open surgical procedures. Most of the users complain of the limited or variable availability of these models in healthcare systems. Remarkably, only 13 participants use 3D models in more than half of their surgical cases.

When stratifying surgeons’ backgrounds (laparoscopists versus robotic surgeons), no statistically significant differences were found in all domains of the survey.

Figure 4 and Figure 5 depict some examples of 3D reconstruction through different softwares (DocDo © 2016–2023; © 2015–2023 Innersight Labs Ltd., London, UK).

## 4. Discussion

During the last decade, 3D models entered urologic surgery, matching the need for customized imaging, one of the major drivers of precision surgery. To date, the main application of 3D models in urology is partial nephrectomy; this is consistent with the outcomes of our survey, in which 85% of participants used those models for PN. Even in the current era, kidney sparing surgery is still demanding since it is adversely impacted by the risk of postoperative hemorrhage in up to 10% of patients and urinary leakage in 1%. Other complications may include injuries to other visceral organs, diaphragm injury, small bowel obstruction, fistulae, and the development of arteriovenous malformations [10]. Surgical preplanning is therefore mandatory to predict the challenging steps of the procedure.

According to the current survey, surgical planning is one of the primary roles of 3D imaging; it involves the creation of a customized surgical roadmap that increases the surgeons’ confidence and guides the decision-making process [19].

Most of the articles published on 3D models for kidney surgery deal with surgical planning. In 2022, Moldovanu et al. published a systematic review addressing the clinical value and applications of 3D virtual reconstruction. A total of 37 articles were found. The number of patients included in the studies ranged from 5 to 157; 23 articles analyzed the impact of 3D models in surgical planning and training, with most of them (14) surveying trainees and patients about their role in understanding renal and tumor anatomy [20].

The role of 3D models in preoperative planning consists of the prediction of surgical strategy. The strategy includes the choice of the approach (i.e., retro/transperitoneal) and the prediction of conversion to radical nephrectomy or to another surgical approach (i.e., from laparoscopic or robotic surgery to open conversion). The 3D models may be used to plan the management of the pedicle, allowing the most appropriate choice between a clampless approach or conventional or selective clamping. They could be used to aid partial nephrectomy or other minimally invasive kidney sparing procedures, such as embolization, cryoablation and radiofrequency [21]. By providing a topographical map of the renal surface and intrarenal anatomy together with the vascularization, 3D models may facilitate avoiding damage to the renal parenchyma and major vessels, while achieving a complete dissection of the tumor.

The measure of the impact of 3D models on surgical planning is usually quantified as the likelihood of a strategy change from those based on 2D imaging [22,23,24,25,26,27,28,29,30,31]. The article from Azhar [6] reports an example of how the planning could change according to 3D models. The author conducted a survey of 100 expert urologists on some real cases of renal surgery shown during an international meeting. Participants were given the 2D CT scans and their corresponding 3D models; they were finally asked to depict their preferred surgical strategy for each case. After viewing the 3D models, the likelihood of a partial approach significantly increased, whereas the choice of radical nephrectomy decreased along with the selection of an open approach. The management of the pedicle turned out to be selective in more cases after viewing the 3D imaging. Based on these outcomes, 3D models resulted in a change in surgical decisions.

The use of 3D models was also found useful in improving the reliability of nephrometric scores in predicting case complexity; some articles addressed the issue, and, generally, scores calculated with 3D images were downgraded in 14 to 67% of cases [19,32,33,34,35]. The occurrence could be explained by the better understanding of the tumor’s depth and contact surface provided by 3D reconstruction compared to 2D images.

Beyond case planning, the use of 3D models for intraoperative navigation is the other application we investigated throughout the survey. It is intended as the real-time consultation of 3D models during surgery, which could be achieved by viewing 3D images on a separate screen (i.e., a smartphone or PC) or inside the Tile Pro robotic console of the Da Vinci (as a picture-in-picture image). A total of 76% of responders found 3D models useful to tailor surgery in a real-time fashion; the occurrence may reflect that several participants were practicing as first surgeons and thus are keener to require “at a glance” consultation of 3D images. Several responders (41%) raised the need for an offline version of the model so that it can be used in case of web connection restrictions.

As far as the clinical benefits of 3D guidance are concerned, until now, only a few articles have evaluated the real advantages with objective and quantifiable measures. As stated in the article from Esperto et al., “good planning doesn’t always mean good surgical outcomes” [36]. The way good planning could translate into clinical advantages may require an analysis and comparison with a control group that should be statistically powered, including issues often underrepresented in the current literature. In retrospective studies, Maddox [37] and Kyung [38] reported the usefulness of 3D model consultation on improving ischemia time, positive margin rate, complication rate and intraoperative blood loss. Fan et al. found those advantages more evident in cases of complex renal mass with a RENAL score ≥ 8 [39].

Kwon Kin et al. performed a prospective case-matched study to compare 40 patients with the application of a 3D-printed transparent kidney model and 40 patients that underwent conventional PN (control group). The endpoint was the difference in console time, which turned out to be reduced by approximately 20% with the 3D tailoring [40].

Apart from the outcomes of these retrospective or nonrandomized analyses, the randomized clinical trial (RCT) from Shirk et al. highlighted the clinical relevance of 3D models in kidney surgery [12]. Ninety-two patients were randomized to receive a partial nephrectomy with or without the use of 3D reconstruction of images, which could be either viewed on the surgeon’s smartphone or in virtual reality using a VR headset. Patients’ covariates and case complexities were similar between groups, and the primary endpoint was operative time. The RCT revealed a difference in estimated blood loss (OR 1.98; 95%CI 1.04–3.78) and length of stay (OR 2.8; 95%CI 1.59–5.14) [12] in favor of surgeries performed with 3D model guidance.

From the current analysis, the positive impact of 3D models on surgical outcomes was perceived by the majority of the participants (63%). Their use for educational purposes was recognized by approximately half of the surveyed, probably reflecting the fact that most of them already completed their training (59%).

Remarkably, only 33% of the surveyed found 3D models as very/highly important for patients’ counseling, meaning that, in real-life practice, they are rarely shown to the patient to improve his/her understanding. This is in contrast with literature findings, in which several authors demonstrated that 3D models can be useful to make the patient understand the challenges of the intervention and improve their compliance, especially the elderly [17,19,41,42,43]. Beyond the relief of presurgical anxiety, 3D models were also found to be able to improve follow-up adherence at 3 years, preventing serious postoperative non-cancer-related complications [44].

Some further considerations may arise. As far as prostate surgery is concerned, preoperative knowledge of local staging is crucial to deciding whether to perform a nerve-sparing surgery or not. Even if affected by a low sensitivity in detecting extracapsular extension [45], mpMRI imaging can play a role in the preoperative and intraoperative setting to display at-a-glance the prostate shape and location of the main tumoral foci. However, the cognitive translation of 2D MRI or ultrasound imaging into the surgical field appears demanding and is globally underused [46,47].

The 3D imaging reconstruction techniques may correspond to an unmet surgical need. Three-dimensional reconstruction, virtual reality and augmented reality can further enhance the impact of real-time imaging guidance during surgery with more realistic and accurate anatomic insights and tumor localization.

A systematic review by Wang [46] published in 2021 analyzed 27 studies to address the impact of 3D printing, virtual reality and AR technologies for PCa procedures; specifically, 22 articles involved the use of such tools in RP. Half of the studies consisted of case series (11) and in some of the cases they referred to the AR, with the 3D model superimposed into the robotic console. In these cases, the advantage relies on the visualization of areas with a possible risk of ECE; however, tissue deformation and automatic tracking of prostate movements still remain the major challenges to be faced. Overall, as concluded by Makary et al. [48] in a review article on the topic, literature evidence supporting the use of image guidance during RP is still scarce. This is consistent with the outcome of the current survey, in which participants barely reported using 3D models before or during prostate surgery.

Another consideration should be finally drafted from the present survey. Despite appreciating these models and endorsing their role as possible game changers, in real-life practice, most of the surveyed (77%) declared to use 3D models in less than 25% of cases. Several factors may account for the occurrence: First, costs may range between USD 1 and 1000 [49], representing a significant limitation to their diffusion into public healthcare systems. Second, in some cases, the development of 3D images can be a time-consuming process, taking from 1.5 h to some days [49]; thus, the time required for 3D reconstruction may not fit the needs of high-volume centers.

The current survey was not devoid of limitations. The most important one is the selection bias, since participants were recruited inside groups highly experienced or somehow exposed to minimally invasive surgery; an inclusion criterion was the current or previous use of 3D models to provide opinions about the issue. Second, the survey was limited to 100 participants to just explore users’ perceptions of the technology. Opposite, an extension of the sample size would have been better highlighting the diffusion of those models among the overall urological community. Finally, all surveys captured the technology at a definite time frame; further advances, i.e., the development of augmented reality or holograms, have not been addressed and may provide different results.

## 5. Conclusions

To our knowledge, this is the largest survey addressing the role of 3D cognitive reconstruction of imaging in urological real-life practice. Overall, 3D models are recognized as useful to improve surgical proficiency, with partial nephrectomy being the procedure most commonly involved. Surgical preplanning and intraoperative consultation are the biggest advantages of 3D reconstruction; however, its educational role (training and patients’ counseling) is recognized to a lesser degree. Costs and time for 3D reconstruction can be considered the main limitations to the diffusion of the technology. Technical refinements—together with a higher availability of the models—may further result in implementing the role of 3D image reconstruction into surgical and clinical daily practice.

## Figures and Tables

**Figure 1 jpm-13-01435-f001:**
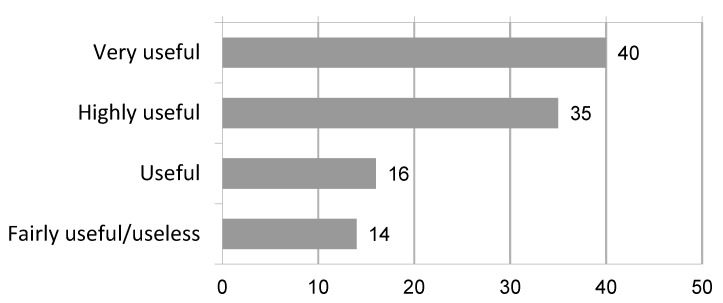
Visualization of replies to the question: “In which setting do you think that 3D imaging reconstruction is more useful? Surgical planning”.

**Figure 2 jpm-13-01435-f002:**
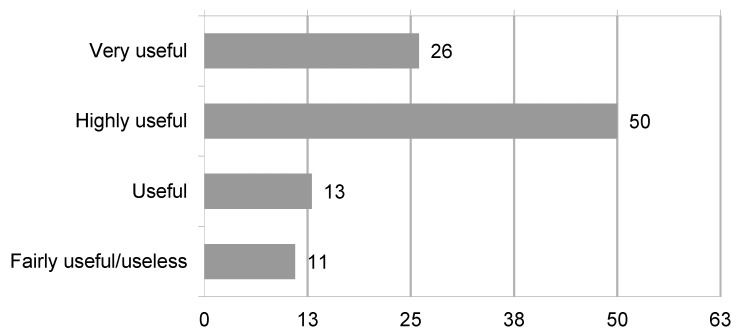
Visualization of replies to the question “In which setting do you think that 3D imaging reconstruction is more useful? Intra-operative consultation/tailoring”.

**Figure 3 jpm-13-01435-f003:**
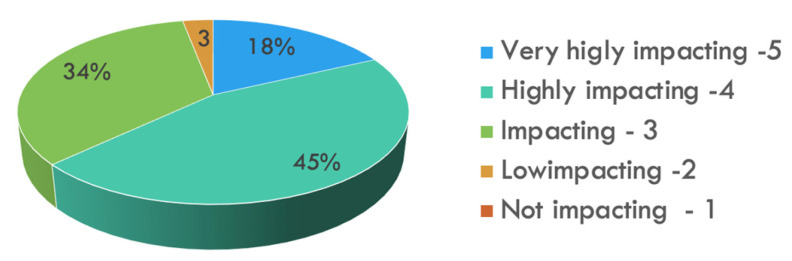
Visualization of replies to the question “How much do you believe 3D imaging use is able to im-prove your surgical outcomes?” (5-Likert scale).

**Figure 4 jpm-13-01435-f004:**
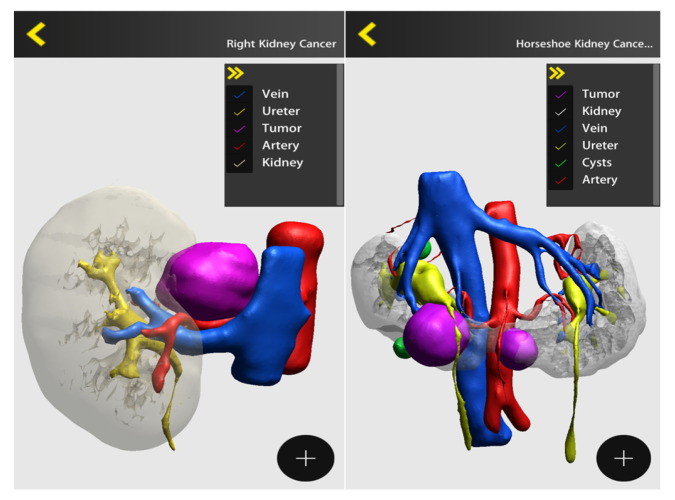
Visualization of 3D images on the smartphone through the DocDo platform (right kidney cancer and horseshoe kidney).

**Figure 5 jpm-13-01435-f005:**
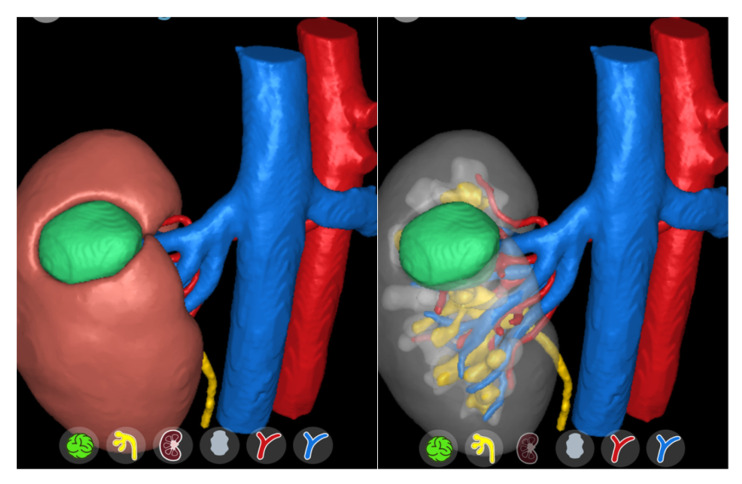
Visualization of 3D images on the smartphone through Innersight platform.

## Data Availability

Not applicable.

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
