# Peer review of "Three-Dimensional Customized Imaging Reconstruction for Urological Surgery: Diffusion and Role in Real-Life Practice from an International Survey"

_jpm, 2023, doi:10.3390/jpm13101435_

Round 1

Reviewer 1 Report

Interesting paper. To improve your survey I recommend to address more if their were differences between open, laparoscopic and robotic surgeons  in the judgement  and use of 3 d reconstructions. Obviously there are technical advantages for robotic surgeons to use 3 d reconstructions. Please add information to your results and discussion

minor:

Results (186): 100 invited , 100 completed (16%) >>> probably 100% is correct

Author Response

We thank the Reviewer for the comments. We tried to improve the paper according to his/her recommendations.

 -Line 318-320: we correct the sentence accordingly

-Line 335-337: we agree with the Reviewer; the paragraph is unclear. We provided insights on the topic and clarify what the importance of 3D models could be.

-Line 364: we added the previous or current use of 3D models as an inclusion criterion in the Materials and Methods section.

-Line 365: We agree with the Reviewer and thank him/her for having highlighted this mistake.       If 100 is the planned number, surely it could not be seen a limitation by itself. We changed the concept in the limitation section.

Reviewer 2 Report

The paper titled "3D Customized Imaging Reconstruction for Urological Surgery: Diffusion and Role in Real-Life Practice from an International Survey" addresses the limited understanding of the usage and perception of three-dimensional (3D) reconstruction models derived from 2D medical imaging within the field of urology. The study aims to shed light on the adoption of such tools among urologists from different backgrounds and geographical origins. The paper suggests that as technical improvements occur and 3D models become more readily available, their role in both surgical and clinical daily practice is likely to expand. This survey highlights the potential benefits of 3D reconstruction models in urological surgery, including improved surgical planning and outcomes, but also underscores the current challenges related to costs and time constraints in their utilization.

Please find below my comments:

-Line 318-320: "The RCT revealed difference in operative time (OR 1; 95%CI 0.37-2.7), estimated blood loss (OR 1.98; 95%CI 1.04-3.78), clamp time (OR 1.60; 95%CI 0.79-3.23) and length of stay (OR 2.8; 95%CI 1.59-5.14) [12] in favor of surgeries performed with 3D model guidance. " As evident from CI crossing the value of 1, clamp time and operative time were not significant in this analysis.  When ORs were estimated using GEE model differences were reported. Please correct. 

-Line 335-337: “As far as prostate surgery is concerned, 3D visualization of prostate shape has been considered crucial so far, since PCa is often infiltrative without a mass effect on surrounding tissue”. I would temper down this sentence since the author reported just promising results of technology in an early stage and coming from low-quality evidence. Moreover, prostate cancer is most of the time multifocal, and the satellite foci might be invisible at the MRI and in consequence, not depictable in a 3D model. Furthermore, why should comprehending the shape of the prostate through a 3D model generated from a prior MRI scan, as opposed to intraoperative awareness, be deemed "crucial" with respect to the thoroughness of tumor eradication? At first glance, is not clear. Please argument.

-Line 364: “an inclusion criterion was the current or previous use of 3D models to provide opinions about the issue.” From the patient selection, no specific criteria were listed as opposite to the one provided in the limitations section, for instance. Please implement.  

-Line 365: “Second, the survey has been upfront 365 limited to 100 participants, and the relatively small sample size could be seen as a limitation.” How was the number predefined? Why, if considered small and limiting the impact of the survey, was chosen as it is?

Author Response

We thank the Reviewer for the comments. We corrected the mistake he/she highlighted (number of responders among those surveyed).

Furthermore, we addressed the issue the Reviewer raised: we tried to analyze if some dissimilarity occurs between laparoscopists and robotic surgeons, but we failed to find any difference. The issue is added in the text.

Round 2

Reviewer 2 Report

Authors addressed all my concerns